# Analysis of Plant Water Transport Mechanism and Water Requirement for Growth Based on the Effect of Thermal Environment

Haolin Lu, Hongfa Sun *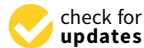 and Jibo Long

College of Civil Engineering and Mechanics, Xiangtan University, Xiangtan 411105, China;
201921002205@smail.xtu.edu.cn (H.L.); hngzlong@xtu.edu.cn (J.L.)
* Correspondence: sunhf1212@xtu.edu.cn

**Abstract:** This paper put forward a model for calculating the water requirements of plants, including a transpiration model, stem water delivery model, and root water uptake model. The results showed that the model had good accuracy. The relative error between simulated values and measured values was 2.09–14.13%. The limiting effects of stem water delivery capacity and root water uptake capacity on plant–water relations were analyzed. When the transpiration rate is large, even if there is enough root water uptake capacity, the limited stem water delivery capacity may affect the plant–water relationship. In order to understand the relationship between a plant and the thermal environment, the effect of the thermal environment on a plant's water requirements was analyzed, and the effect of air temperature was obvious. Under the simulated condition, when the air temperature increased from 0 °C to 40 °C, the water requirement of an apple tree increased from 0.0134 L/h to 33.8 L/h.

**Keywords:** plant water requirement; thermal environment; plant–water relation; stem water delivery

## 1. Introduction

Global warming has become an important issue affecting the sustainable development of human society. Therefore, carbon neutralization has been contributing a lot to the sustainable development of society. At present, plants' carbon storage has become an economic and effective method to reduce the content of carbon dioxide in the atmosphere. Therefore, creating a suitable growth environment for plants, and making full use of land resources have become very important problems to be solved in the development of plant carbon storage technology [1,2].

Many factors can affect the growth environment of a plant, which include sunlight, water, air temperature, humidity, and wind velocity [3]. When the canopy is affected by environmental changes, transpiration pull will be generated in the plant, transmitting water from the root system to the leaves through the stem. When a plant lacks water directly or indirectly, it may lose vitality and wilt. Therefore, it is of great significance to accurately model the plant's water requirements to predict whether it will lose vitality [4]. The modelling of transpiration has been a popular research area for scholars. The current studies on transpiration mainly focus on four levels: single leaf level, whole plant level, stand level, and regional level. The modelling of single leaf transpiration and whole plant transpiration are the basis for calculating the plants' water requirement of stands or regional vegetation [5]. Compared with the whole plant transpiration model, the single leaf transpiration model can accurately predict the transpiration. Based on the aerodynamics theory, Yuan [6] proposed a single leaf transpiration model that needed less environmental data, and the model could also obtain precise results. However, due to the limitation of the single leaf transpiration model, it may be difficult for it to be used to model the transpiration of the whole plant. The plant adjusts the intensity of physiological activities such as transpiration through stomatal movement [7], and the stomata could be affected by

many factors. Research also showed that the stomata respond to both climate factors and soil moisture changes [8], and the whole plant–water relationship could be affected by the degree of stomatal movement. However, under severe thermal environmental conditions, stomatal control would not be enough to effectively prevent the water loss of a plant [9].

As the channel for the water transport in a plant, the stem water delivery capacity is affected by the anatomical features of the xylem and the physical properties of sap in the plant [10,11]. At present, scholars have conducted much relevant research on sap movement in plants [10,12], and many great results have been found. Because of the invariable anatomical features of the xylem, there may exist a maximum value of the stem water delivery rate. As one of the organs used by the plant to absorb water, the root can be affected by transpiration, soil water potential, and root length density [13,14]. The root water uptake model can be divided into a micro model and macro model [15], and the micro model was proposed by Gardner first. However, due to the complex system of the root, it has been difficult to describe its water uptake process.

Therefore, the plant water transport mechanism was studied in this paper; a plant water requirement model was established, which included a transpiration model, stem water delivery model, and root water uptake model. Our objective is to analyze the limiting effect of the stem and the influence of the thermal environment on plants' water requirements. The results contribute to the ideas for the irrigation strategies of plants, the rational distribution of water resources, and the efficient utilization of the limited land resources. In some aspects, the model established in this paper could also promote the development of plant carbon storage technology.

## 2. Methods

The physical model of plant's water requirements is shown in Figure 1.

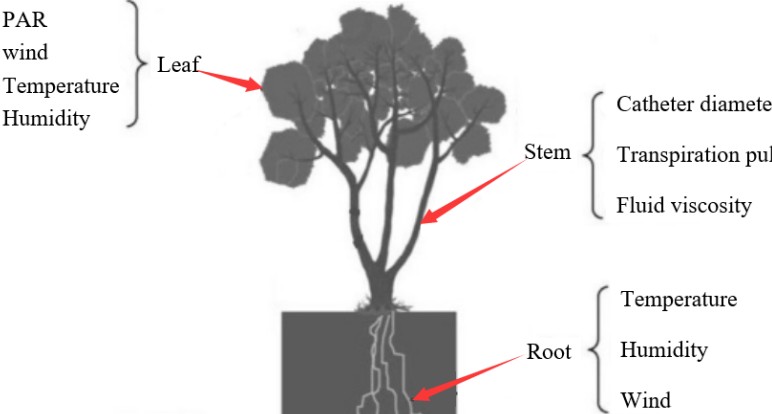

**Figure 1.** The plant physical model.

The plant water requirement model is composed of transpiration model, stem water delivery model, and root water uptake model. In this model, thermal environmental factors are used as input parameters, including solar radiation, air temperature, air relative humidity, and wind velocity.

### 2.1. Transpiration Model

The complex physiological functions of plants make it difficult to calculate transpiration. In earlier studies, people chose to simplify the canopy description to solve this problem. The single leaf transpiration model is simple in form, and the relation between leaf activities and thermal environment can be described well. Therefore, it was possible to obtain accurate results through the single leaf transpiration model and extend it to the whole plant scale [16].

The main transpiration organ is the stoma, from which water can be diffused to the atmosphere by plant. The power of transpiration is the concentration difference between the saturated vapor around the stomata and that in the ambient air. The resistances are stomatal resistance and boundary layer resistance [6]. Therefore, the transpiration rate of single leaf can be expressed as follows

$$q_1 = \frac{C_{leaf} - C_{air}}{r_s + r_a} \tag{1}$$

where, $C_{leaf}$ is leaf water vapor concentration, (kg/m$^3$). $C_{air}$ is air water vapor concentration, (kg/m$^3$). $r_s$ is stomatal resistance, (s/cm). $r_a$ is boundary layer resistance.

According to Goff formula, the partial pressure of saturated vapor is a single valued function of temperature

$$p_s = 1026 \exp[52.67 - 6790/(273 + T) - 5.03 \ln(273 + T)] \tag{2}$$

where, $P_s$ is saturated vapor pressure, (hPa). $T$ is temperature, (°C).

Therefore, the numerator in Equation (1) can be written as

$$C_{leaf} - C_{air} = 7.26 \times 10^{-6}(p_{s,leaf} - RH \cdot p_{s,air}) \tag{3}$$

where, $P_{s,leaf}$ is leaf vapor pressure, (hPa). $RH$ is relative humidity, (%). $P_{s,air}$ is air vapor pressure, (hPa).

The temperature of leaf could be calculated from empirical formula [17]:

$$\begin{aligned} T_{leaf} = & v^{0.04607} LAI^{-0.03198} w^{0.00768} T_{air} - (1 - RH) \cdot v^{-0.55678} LAI^{-0.39479} T_{air} \\ & + 0.10827 \varphi u^{-1.01464} LAI^{-0.33403} w^{-0.09859} T_{air}^{-0.59998} R_s \end{aligned} \tag{4}$$

where, $T_{leaf}$ is leaf temperature, (°C). $v$ is wind speed, (m/s). $LAI$ is leaf area index. $w$ is leaf width, (m). $T_{air}$ is air temperature, (°C). $\varphi$ is leaf coverage rate. $R_s$ is solar radiation value, (W/m$^2$).

The resistance of boundary layer could be calculated by the following formula. When the wind velocity is less than 0.1 m/s

$$r_a = 840 \left(\frac{d}{\left|T_{leaf} - T_{air}\right|}\right)^{0.25} \tag{5}$$

and when the wind velocity is larger than 0.1 m/s

$$r_a = 220 \frac{d^{0.2}}{v^{0.8}} \tag{6}$$

It was found that the stomatal resistance model proposed by Jarvis [18] directly described the effect of thermal environment on plant's water requirements. In order to reflect on the effect of thermal environment on plant, the Jarvis model was cited to calculate the stomatal resistance in this paper. The stomatal conductance model proposed by Jarvis is as follows

$$g_s = g_{s,max} \cdot f_1(PAR) f_2(VPD) f_3(T) \tag{7}$$

where, $g_s$ is stomatal conductance value. $g_{s,max}$ is maximum stomatal conductance value. $VPD$ is water vapor pressure difference between leaf and air.

The response function of stomatal conductance to effective solar radiation is [19]

$$f_1(PAR) = PAR/(a_1 + PAR) \tag{8}$$

The response function of stomatal conductance to atmospheric vapor pressure deficits is [20]

$$f_2(VPD) = 1/(a_2 + VPD) \tag{9}$$

The method for calculating atmospheric vapor pressure deficits is as follows [21]

$$VPD = 0.611 \exp(17.4 \cdot \frac{T_{air}}{239 + T_{air}}) - 0.611 \exp(17.4 \cdot \frac{T_s}{239 + T_s}) - 0.067(T_{air} - T_s) \tag{10}$$

The response of stomatal conductance to leaf temperature is as follows [22]

$$f_3(T) = a_3 T^3 \tag{11}$$

Research showed [23] that the empirical parameters in the above equations played important roles in modelling the plant water requirement. In addition, this is not an entirely new finding; other studies [8,24] have provided additional validation that the stomata also respond to factors other than those in the above equations.

The conversion relationship between stomatal resistance and stomatal conductance is as follows

$$r_s = 12.1875 \frac{P}{T} g_s \tag{12}$$

where, $P$ is atmospheric pressure, (hPa).

From Equations (1)–(12), leaf transpiration rate could be described by the following Equation

$$q_1 = \begin{cases} \dfrac{7.26 \times 10^{-6}(p_{s,leaf} - RH \cdot p_{s,air})}{840 \left( \dfrac{d}{|T_{leaf} - T_{air}|} \right)^{0.25} + r_s} & v \leq 1\,\text{m/s} \\[4ex] \dfrac{7.26 \times 10^{-6}(p_{s,leaf} - RH \cdot p_{s,air})}{220 \dfrac{d^{0.2}}{v^{0.8}} + r_s} & v > 1\,\text{m/s} \end{cases} \tag{13}$$

The relationship between leaf transpiration and whole plant transpiration could be established by leaf area index [25]. Therefore, transpiration rate of the whole plant can be written as follows

$$Q_1 = \frac{7.26 \times 10^{-6}(p_{s,leaf} - RH \cdot p_{s,air})}{r_s + r_a} \cdot 3600 \cdot LAI \cdot S \tag{14}$$

where, $S$ is planting area, (m$^2$).

### 2.2. Stem Water Delivery Model

As the channel connecting canopy and root, the stem plays an important role in the process of plant water transport. The stem water delivery rate could be affected by conduit density, diameter, filling degree, and internal characteristics [10]. According to Hagen equation, under a steady condition of atmospheric pressure, the stem water delivery rate could be directly proportional to the fourth power of the xylem catheter diameter [26]. Studies have shown that stem vessels could be integrated through stem diameter and correction coefficient [9,26]. Therefore, the stem water delivery rate could be calculated by the following formula [10]

$$Q_2 = \frac{\pi r^4 \Delta P}{8 \eta L} \tag{15}$$

where, $r$ is duct radius, (m). $\eta$ is dynamic viscosity of water, (Pa·s).

Equation $\Delta P / L$ represents the pressure gradient at both ends of xylem. The calculation method is as follows

$$\frac{\Delta P}{L} = P_{air} \cdot S + \rho_{liquid} \cdot L \cdot S \cdot g \tag{16}$$

where, $P_{air}$ is atmospheric pressure; $S$ is xylem diameter; $\rho_{liquid}$ is density of liquid in xylem; $L$ is xylem length; $g$ is gravitational acceleration.

Recent studies have shown that the stem water delivery capacity declines with the increase in the transpiration rate [27]. Therefore, this phenomenon should be considered when calculating the stem water delivery rate of plant.

### 2.3. Root Water Uptake Model

In the existing research, a complete set theory, which can calculate the root water uptake rate by layer, has been established. However, the traditional root water uptake models may focus on the changes in water uptake at different horizontal sections. When the stem water delivery capacity meets the plant water requirement, it can be assumed that the root water uptake rate is equal to the transpiration rate. Therefore, the root water uptake rate can be expressed by formula (14)

$$Q_3 = \frac{7.26 \times 10^{-6}(p_{s,leaf} - RH \cdot p_{s,air})}{r_s + r_a} \cdot 3600 \cdot LAI \cdot S \tag{17}$$

Furthermore, in some growth periods of plant, transpiration rate and root water uptake rate may not be equal. In order to illustrate the effect of thermal environment on plant–water relation. The taproot sap movement model could be used to model the water uptake rate of the root [28]

$$q_r = a_4 T_{air}{}^{\alpha}(b_1 RH^2 + b_2 RH + b_3)(c_1 v + c_2) \tag{18}$$

where, $a_4$, $b_1$, $b_2$, $b_3$, $c_1$, $c_2$ are empirical coefficients.

The plant water uptake capacity of the whole root system can be about 6 times that of the main root [28], therefore the method of calculating the water uptake rate of the whole root system can be expressed as follows

$$Q_3 = K_2 \cdot a_4 T_{air}{}^{\alpha}(b_1 RH^2 + b_2 RH + b_3)(c_1 v + c_2) \tag{19}$$

where, $K_2$ is empirical coefficient.

In order to reflect the relationship between plant water requirement and thermal environment, the root water uptake model was established based on thermal environmental factors. However, factors affecting root water uptake are very complex, so it is essential to modify the model with empirical parameters.

### 2.4. Plant Water Requirement Model

According to the analysis in Sections 2.1–2.3, when the root water status is sufficient and the stem water delivery capacity can meet the water requirements of transpiration, the plant water requirement can be expressed as follows

$$Q = \frac{7.26 \times 10^{-6}(p_{s,leaf} - RH \cdot p_{s,air})}{r_s + r_a} \cdot 3600 \cdot k_1 \cdot LAI \cdot S \tag{20}$$

When the stem water delivery capacity meets the transpiration water requirement but the root water status is insufficient, the plant may lose vitality and wither. At this time, the plant water requirement can be expressed as follows

$$Q = k_2 \cdot a_4 T_{air}{}^{\alpha}(b_1 RH^2 + b_2 RH + b_3)(c_1 v + c_2) \tag{21}$$

Furthermore, when the stem water delivery rate is far less than the water requirement of transpiration, the plant may wither because of water shortage. At this time, the plant water requirement could be much larger than its stem water delivery rate

$$Q = \frac{\pi r^4 \Delta P}{8 \eta L} \tag{22}$$

From the analysis made above, the closing degree of stomata have a direct impact on plant water requirement. According to the stomatal conductance model shown in Equation (7), air temperature could affect the stomatal conductance powerfully, and solar radiation intensity is an important factor affecting air temperature. Therefore, it could be effective to regulate the thermal environment to maintain the growth of plant. The maximum solar radiation that the plant is able to withstand can be obtained from Equations (7) and (20)

$$\frac{PAR}{PAR + a_1} = \frac{k_1 - Q_2}{k_2} \tag{23}$$

Among them,

$$\begin{cases} k_1 = 7.26 \times 10^{-6}(p_{s,leaf} - RH \cdot p_{s,air}) \cdot 3600 \cdot LAI \cdot S \\ k_2 = g_{s,max} \cdot f_2(VPD)f_3(T) \end{cases} \tag{24}$$

## 3. Results and Discussion

### 3.1. Model Validation

#### 3.1.1. Materials and Methods

In recent years, the thermal pulse generator has become a popular instrument to measure the liquid velocity in the vessel of the xylem. Taking an apple tree as the calculation object, the plant water requirement model was validated by comparing the simulated results with the measured stem sap flow rate. Gong [28] made great measurements of the hourly transpiration rate of apple tree. The results were measured from 31 July to 4 August in 2005 using a thermal pulse generator. Two thermal pulse probes were installed at different heights on the east and west sides of the middle of the trunk, and the transpiration rate was measured with a SF100 stem sap flow meter. The diameter of the experimental object was 113 mm, the crown height was 2.35 m, and the width was 2.78 m. The environmental data were measured by the automatic weather station of the experimental station. The experiment was conducted in Northwest University of agriculture and forestry science and technology.

The main environmental factor values are shown in Figures 2–5.

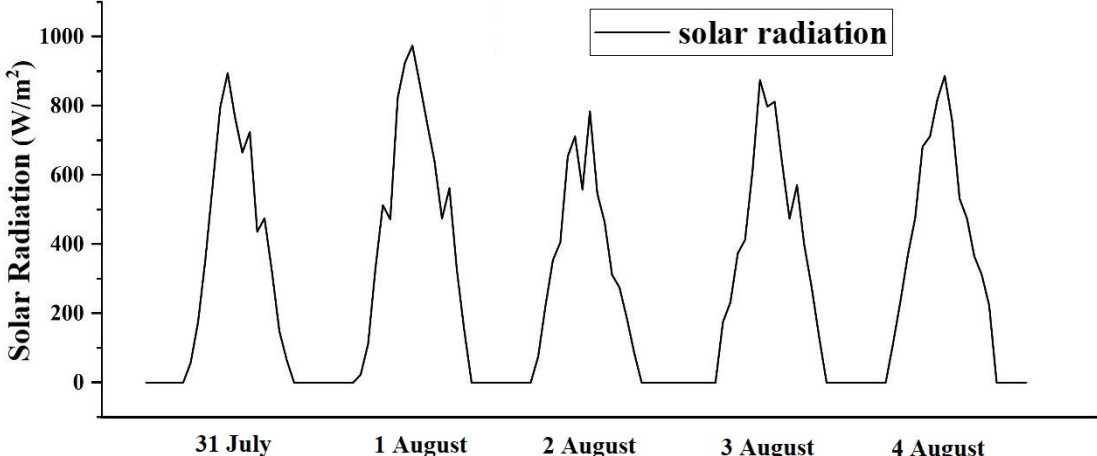

**Figure 2.** Variation in solar radiation.

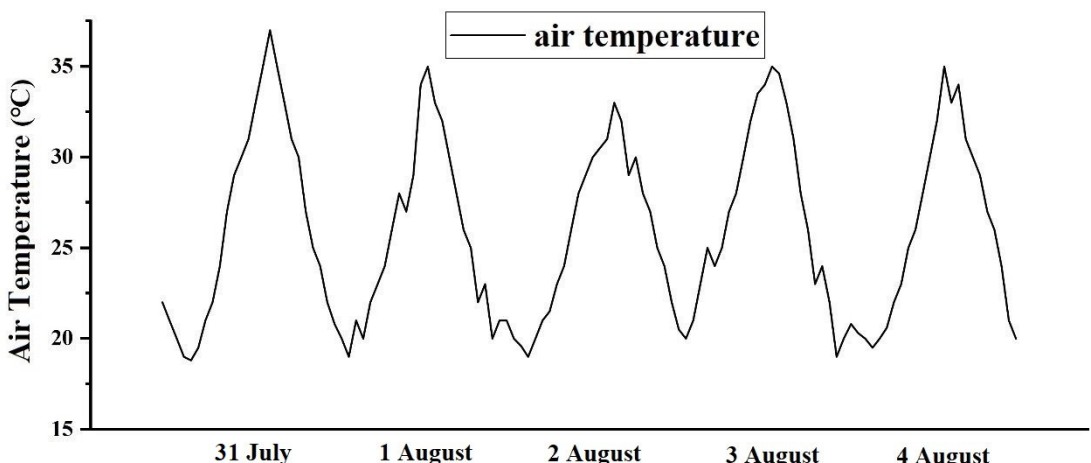

**Figure 3.** Variation in air temperature.

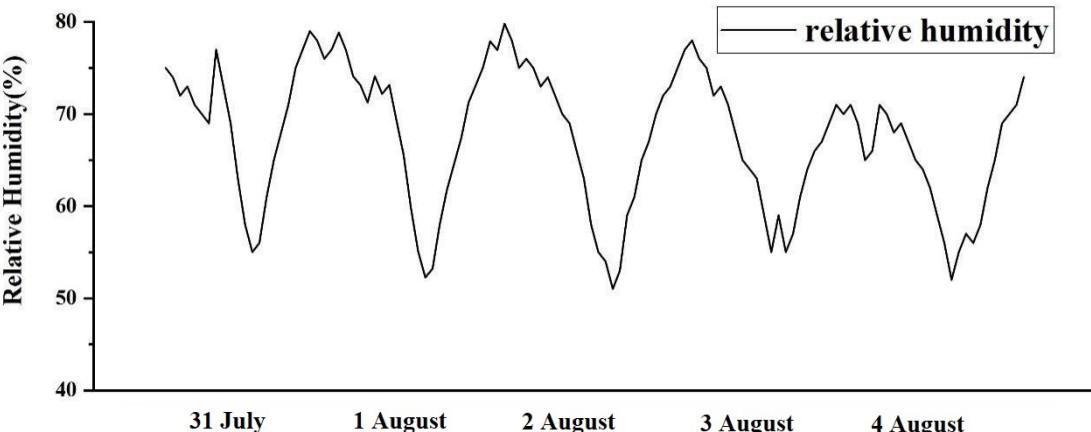

**Figure 4.** Variation in air relative humidity.

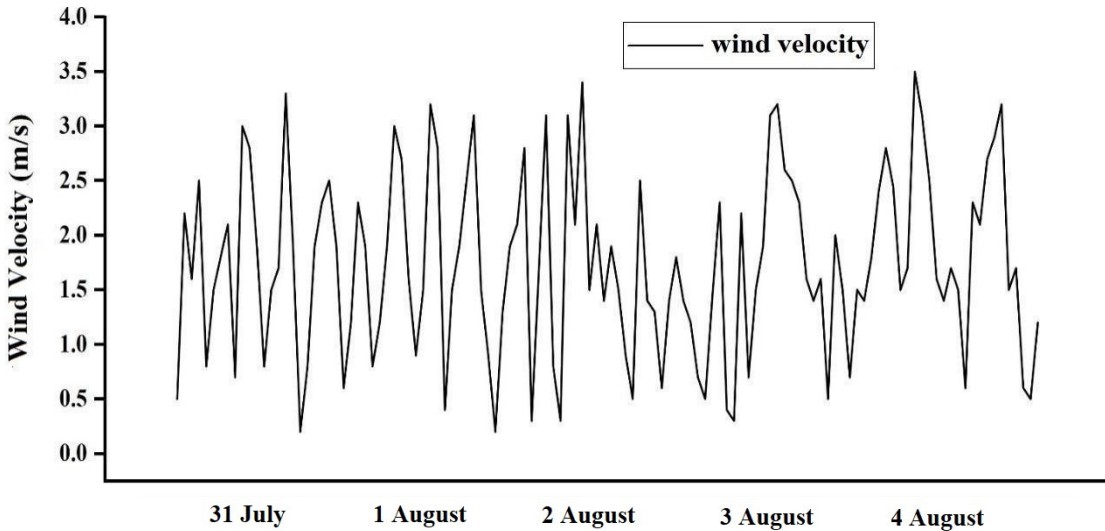

**Figure 5.** Variation in wind velocity.

The hourly variations in solar radiation intensity, air temperature, air relative humidity, and wind velocity are described in Figures 2–5. From Figures 2–5, the variation range of daytime solar radiation intensity was 57–925 W/m², the variation range of air temperature

was 17.9–36.8 °C, the variation range of air relative humidity was 51–79%, and the variation range of wind velocity was 0.2–3.6 m/s.

### 3.1.2. Validation

The empirical parameters in the stomatal resistance model and taproot sap flow model were fitted with measured environmental parameters. The fitting values are shown in Table 1.

**Table 1.** Empirical parameters in stomatal resistance model and taproot sap flow model.

| $a_1$ | $a_2$ | $a_3$ | $a_4$ | $b_1$ | $b_2$ | $b_3$ | $c_1$ | $c_2$ |
|-------|-------|-------|-------|-------|-------|-------|-------|-------|
| 88.55 | 8.98 | 0.0025 | 2.36475 | 0.025357 | −2.89317 | 133.9857 | 0.00658 | 0.003744 |

The correlation coefficient of stomatal resistance model was R = 0.83, and the correlation coefficient of taproot sap flow model was R = 0.88.

Based on the above measured data, the transpiration rate of the apple tree could be calculated by Equations (19)–(21). The comparison between the simulated values and the measured values is shown in Figure 6.

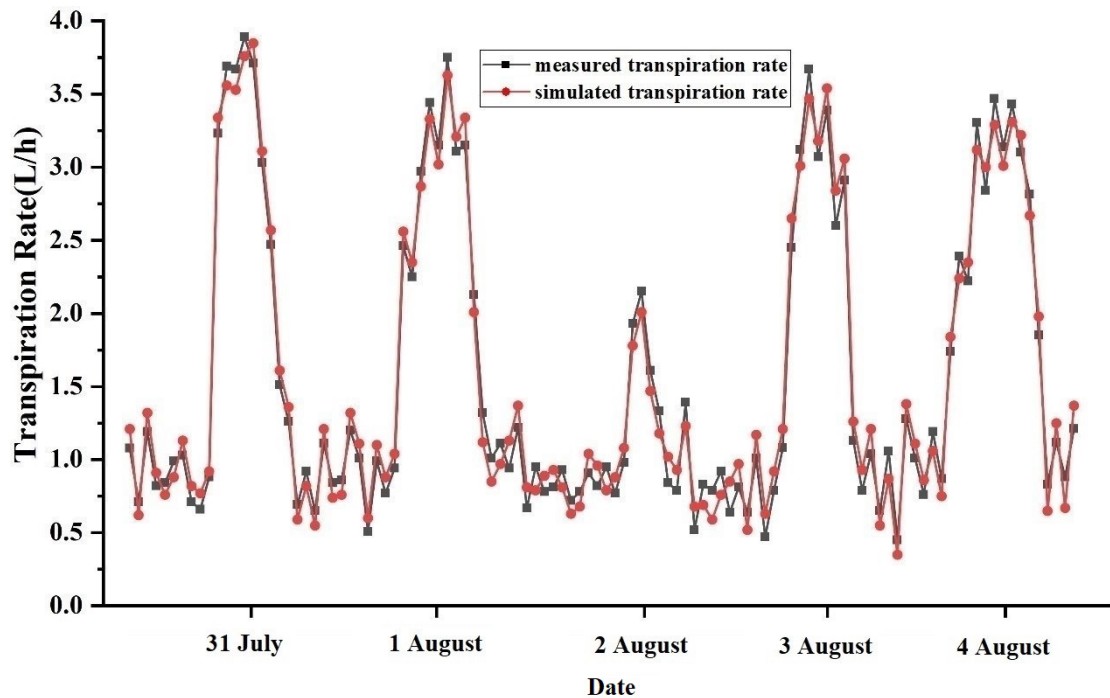

**Figure 6.** Verification of transpiration rate model.

As shown in Figure 6, the water requirement of the apple tree on 2 August is at a relatively low value, which may be due to the low temperature and low solar radiation during this period, resulting in the increase in stomatal resistance and slowing down of transpiration.

In order to validate the accuracy of the model, the absolute error and relative error of the model were calculated. The absolute error range was 0.08–0.22 L/h, and the relative error range was 2.09–14.13%. Furthermore, the correlation coefficient of the plant water requirement model was R = 0.87. The results indicated that the simulated values of the transpiration rate model were in good agreement with the measured values.

### 3.2. Discussion

From the analysis of the Section 2, it can be concluded that the thermal environmental factors affecting plants' water requirements are solar radiation, air temperature, air humid-

ity, and wind velocity. In this section, the effect of these factors on plant–water relations, leaf water status, sap movement of the plant, and stomatal activities are further discussed.

### 3.2.1. Effect of Solar Radiation

Assuming that the leaf diameter of an apple tree is 0.05 m, the stem diameter is 0.15 m, and the height is 2 m. The stem water delivery rate can be calculated as 16.21 L/h from Equation (15).

Assuming that the air relative humidity is 50%, the wind velocity is 1 m/s, and the air temperatures are 10 °C, 20 °C, 30 °C, and 40 °C, when the solar radiation intensity changes in the range of 100–1000 W/m², the plant's water requirements can be obtained as shown in Figure 7.

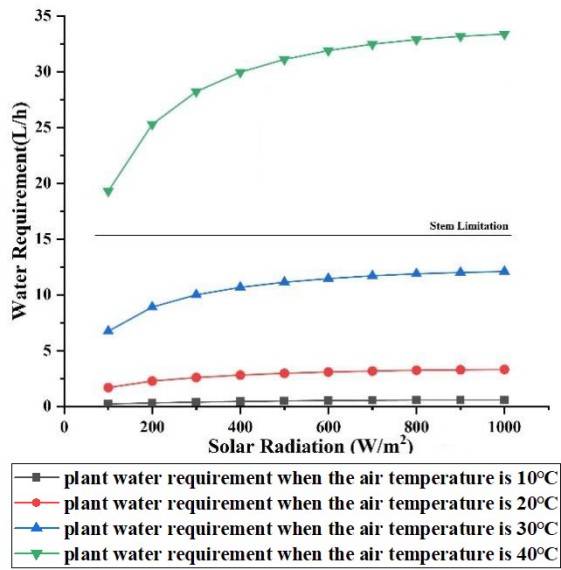

**Figure 7.** Relationship between minimum water requirement and effective solar radiation.

As shown in Figure 7, the plant's water requirements increased with the increase in solar radiation. When the solar radiation increased from 100 W/m² to 1000 W/m², and the variation range of air temperature was 10 °C–40 °C, the plant's water requirements increased from 0.35 L/h to 14.08 L/h. Furthermore, when the air temperature was 40 °C, the stem water delivery capacity could not meet the water requirement. At this point, stomatal regulation may not improve the plant's vitality effectively.

According to Wang's theory, the stem water delivery capacity is affected by the physiological characteristics of xylem and the hydrodynamic viscosity of irrigation water. Xylem's physiological characteristics such as vessel diameter, vessel number, and vessel size can affect the plant's sap movement, and the sap movement can affect plant–water relations [12]. Therefore, when temperature is high enough, even if the water status of the root is sufficient, the plant's water requirements may not be met due to the limiting effect of the stem.

### 3.2.2. Effect of Air Temperature

Assuming that the relative humidity of air is 50%, the wind velocity is 1 m/s, and the solar radiations are 100 W/m², 300 W/m², 500 W/m², and 1000 W/m², when the air temperature changes in the range of 2–40 °C, the plant's water requirement can be obtained as shown in Figure 8.

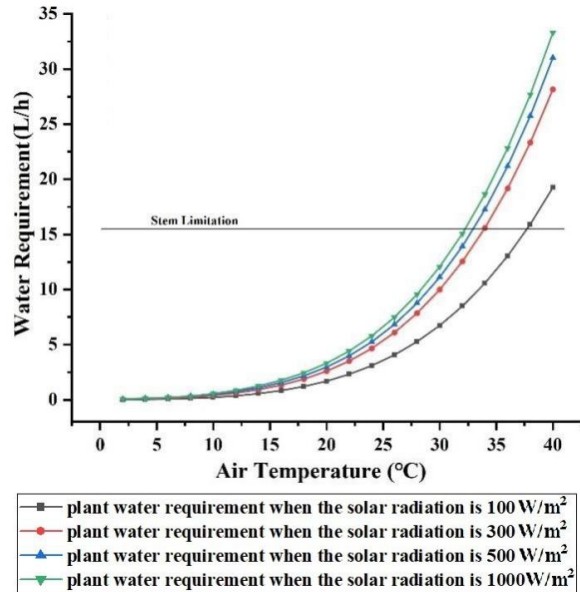

**Figure 8.** Relationship between water requirement and air temperature.

As shown in Figure 8, the plant's water requirement exponentially increased with the increase in air temperature. Therefore, when the environment conditions are serious, the ability of stomata to regulate leaf water status may be insufficient. The reason could be that under high temperature stress, the regulation ability of stomata could reach the limit. A study showed that heat or drought affects the water conserving capacity of leaves [7]. It can also be seen from Figures 7 and 8 that heat or drought affects the plant's water requirement.

It can also be seen from Figure 8 that when the temperature rose to a certain value, the stem water delivery capacity could not meet the water requirements of the plant. Furthermore, the corresponding temperature value of this point decreased with the increase in solar radiation.

### 3.2.3. Effect of Relative Humidity

Assuming that the air temperature is 25 °C, the wind velocity is 1 m/s, and the solar radiations are 100 W/m$^2$, 300 W/m$^2$, 500 W/m$^2$, and 1000 W/m$^2$. When the air relative humidity changes in the range of 30–70%, the plant's water requirements can be obtained as shown in Figure 9.

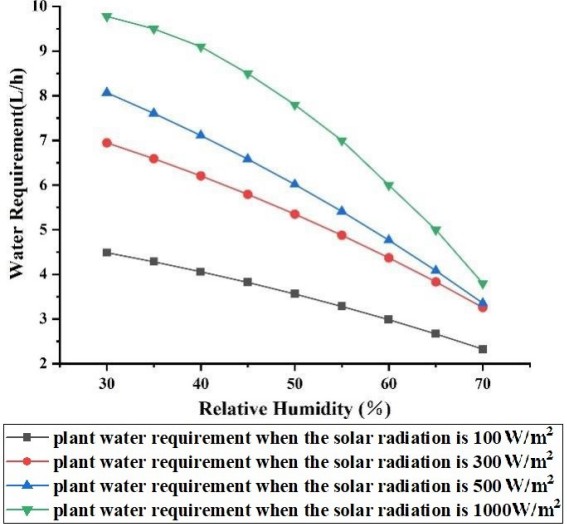

**Figure 9.** Relationship between water requirement and relative humidity.

As shown in Figure 9, the plant's water requirements decreased with the increase in relative humidity. The reason could be that with the increase in air relative humidity, the vapor in the air around the plant decreased, resulting in the increase in the boundary resistance. In addition, with the increase in relative humidity, the water requirements of the plant decreased. It can also be seen from the figure that when the air relative humidity increased from 30% to 70%, the variation range of the plant's water requirements under different solar radiation intensities were quite different. Furthermore, with the increase in radiation value, the variation range of the plant's water requirements increased.

### 3.2.4. Effect of Wind Velocity

Assuming that the effective solar radiation is 500 W/m$^2$, the relative humidity is 50%, and the air temperatures are 10 °C, 20 °C, 30 °C, and 40 °C, when the wind velocity changes in the range of 0.2–3 m/s, the plant's water requirements can be obtained as shown in Figure 10.

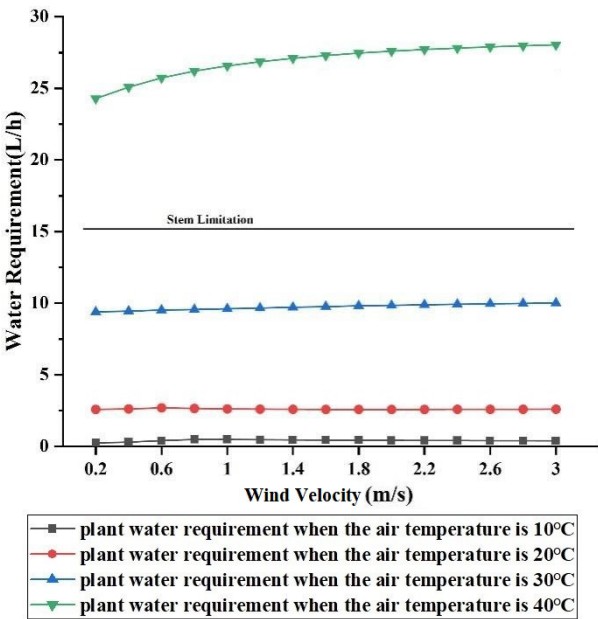

**Figure 10.** Relationship between water requirement and wind velocity.

As shown in Figure 10, when the wind velocity changed in the range of 0.2–3 m/s, the plant water requirement increased with the increase in wind velocity, and the trend was not obvious. The reason could be that the increase in wind velocity led to the decrease in boundary layer resistance, and the plant's water requirements increased. It can also be seen from the figure that when the air temperature was 10 °C and 20 °C, the change in the plant's water requirements relative to wind velocity first increased and then decreased.

### 3.2.5. Effect of Multiple Parameters

According to the analysis of Figures 7–10, the changes in solar radiation and air temperature had significant effects on the plant's water requirements. Assuming that the absolute moisture content is constant, and the wind velocity is 1 m/s, in order to further illustrate the effect of the thermal environment on plants' water requirements, the variation trend of the water requirement with temperature and solar radiation under the conditions of a cold season (5–20 °C) and hot season (20–40 °C) were simulated as shown in Figure 11.

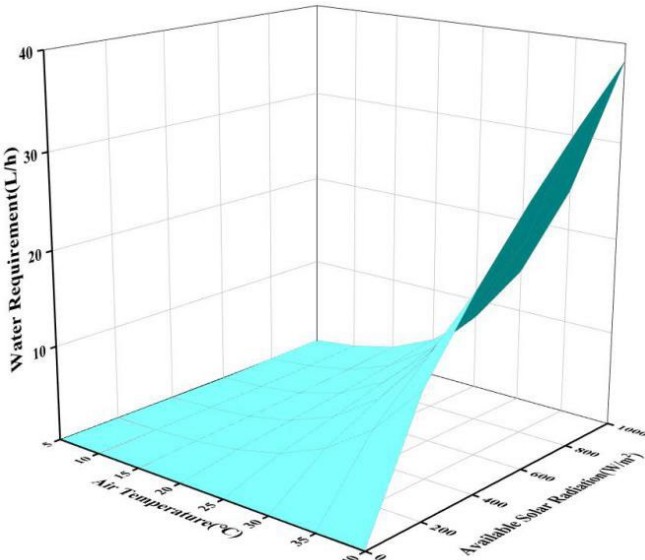

**Figure 11.** Effects of multiple parameters on plant water requirement.

As shown in Figure 11, the plant's water requirements increased with the increase in solar radiation and air temperature, and the change in air temperature had a more significant effect on the water requirement. Furthermore, it can be obtained from Equations (15) and (16) that, under a simulated condition, the stem's water delivery rate of the apple tree was 16.21 L/h, which cannot meet the apple tree's water requirements under all meteorological conditions as shown in Figure 11. This phenomenon showed that under severe thermal conditions, stomatal activities could not fully regulate the water transport process of a plant, and the thermal environmental factors have significant effects on plant–water relations. Therefore, regulation of the thermal environment may play a positive role in maintaining the vitality of plants.

These findings could provide a theoretical basis for the regulation strategy of the agricultural environment. When stomatal regulation cannot regulate the plant–water relationship effectively, or the stem water delivery capacity is insufficient, the effect of appropriate regulation of the air temperature may be positive. In previous studies, scholars have deeply analyzed the response mechanism of plants to environmental factors [29,30]. Based on these great findings, the plant water requirement model was established in this paper, the stem water delivery model was optimized, and the root water uptake model was summarized based on the thermal environment. In order to reflect the relationship between the thermal environment and plants' water requirements, the stomatal model and root water uptake model were established based on thermal environment factors. However, the factors affecting stomatal movement and root water uptake are very complex; the empirical parameters are essential to modify these models. Furthermore, it can be found from the discussion above that severe thermal environments may lead to the loss of plant vitality, and the limiting effect of the stem may be an important reason for this phenomenon. Therefore, compared with previous studies, the model established in this paper can describe the plant's water transport process macroscopically, and may reflect the effect of the thermal environment on plants' water requirements in some other aspects.

## 4. Application and Analysis

In this section, the summer meteorological data of Lintong and Turpan were used as environmental parameters to describe the variation in the apple tree's water requirements, and the irrigation strategies and cooling effects of plants were analyzed.

### 4.1. The Characteristics of the Study Plots

Lintong is located in the east of Guanzhong Plain, China. The annual average air temperature in Lintong was 13.5 °C. During the year, the highest air temperature was 36.8 °C, and the lowest air temperature was −0.9 °C. In summer, the air temperature is high and the solar radiation resources are abundant. In winter, the air temperature is low, the solar radiation resources are relatively scarce, and frozen soil sometimes occurs.

Turpan is located in the central part of Xinjiang Uygur Autonomous Region, China. The climate of Turpan is a continental warm temperate desert climate, with sufficient sunshine and abundant solar energy resources. The annual sunshine hours reached 3200 h, and the annual average air temperature was 13.9 °C, of which the air temperature was higher than 35 °C for more than 100 days. During the year, the highest air temperature reached 49.6 °C, and the lowest air temperature was −28.7 °C.

### 4.2. Analysis on Irrigation Strategy and Thermal Environment Regulation Ability of Plant

The environmental data were obtained from the website of China Meteorological Administration. The data of solar radiation intensity and air temperature in Lintong and Turpan are shown in Figure 12:

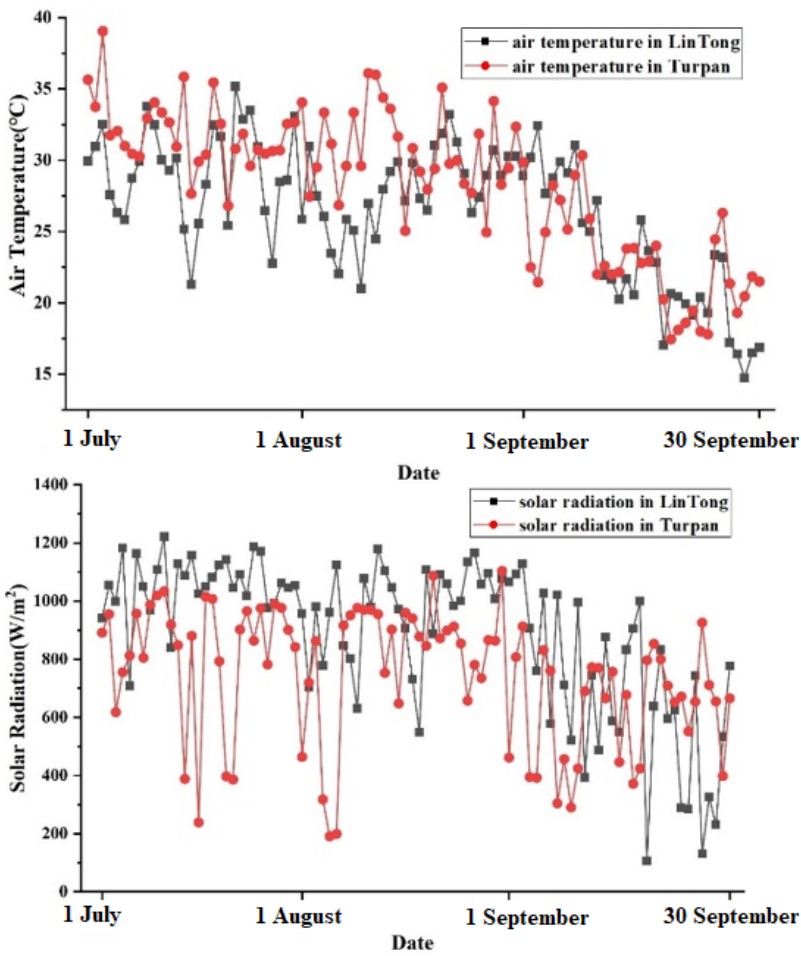

**Figure 12.** Air temperature and solar radiation intensity of the Lintong and Turpan.

As shown in Figure 12, in Lintong, the air temperature varied from 14.73 °C to 35.20 °C, and the solar radiation varied from 113 W/m$^2$ to 1274 W/m$^2$. The air temperature of Turpan varied from 17.45 °C to 39.05 °C, and the solar radiation varied from 172 W/m$^2$ to 1018 W/m$^2$. According to the environment data, the daily water requirement of an apple tree can be obtained as shown in Figure 13.

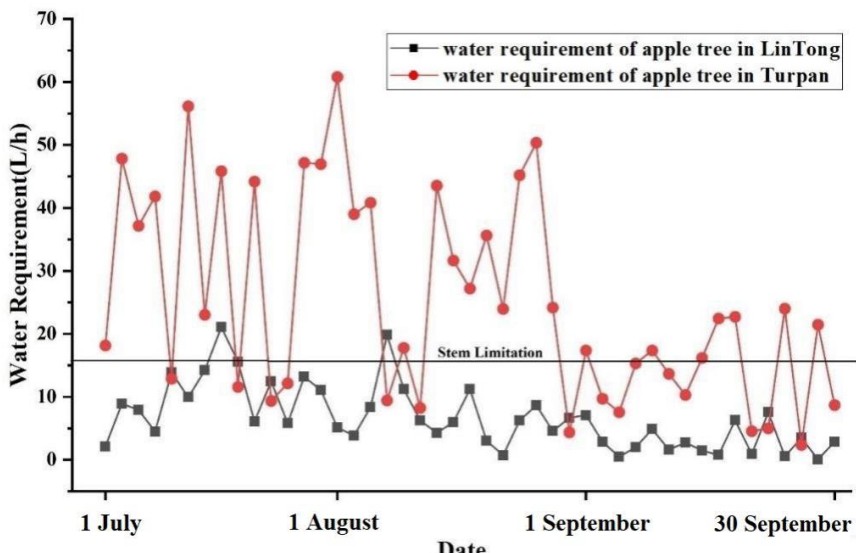

**Figure 13.** Variation in water requirement of apple tree in Lintong and Turpan.

As shown in Figure 13, the daily variation in the apple tree's water requirements was similar to that of the air temperature and solar radiation. In Lintong, the maximum value of water requirement was 21.9 L/h, and its overall trend was relatively gentle. In Turpan, the value was 62.38 L/h, and the plant water requirement changed greatly. It can also be seen from Figure 10 that in Lintong, the plant's water requirements on 17 July and 7 August were greater than the stem water delivery rate. In Turpan, the stem water delivery capacity could not meet the water requirements under most conditions.

The average value of the plant's water requirements can be calculated in a month. The calculation day with the water requirement closest to the average value was defined as the typical day. After accumulating the hourly water requirement, the total plant water requirement could be multiplied by the calculation days of each month. In Lintong, according to Figure 13, the average daily water requirement in summer was 10.12 L/h, 7.19 L/h, and 2.56 L/h, respectively, and the typical days were 15 July, 17 August, and 16 September. In Turpan, the average daily water requirement was 34.34 L/h, 27.92 L/h, and 13.41 L/h, respectively, and the typical days were 7 July, 21 August, and 15 September. The hourly water requirements of an apple tree in Lintong and Turpan on a typical day are shown in Figure 14:

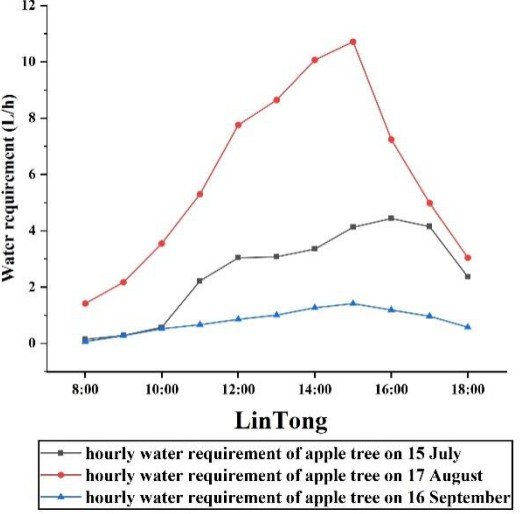

**Figure 14.** *Cont.*

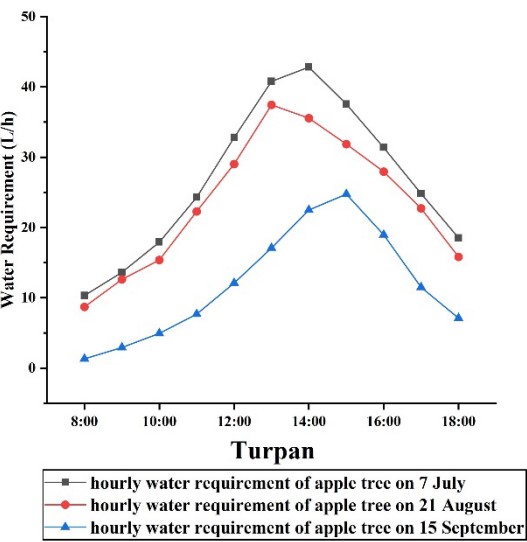

**Figure 14.** Variation in hourly water requirements of apple tree in Lintong and Turpan.

As shown in Figure 14, in Lintong, the apple tree's water requirements on a typical day showed a single peak change law of first rising and then falling, in which the peak value of 10.72 L/h was on 17 August. Furthermore, the daily plant water requirement on 17 August was higher than those on the other days. The reason could be that the temperature and radiation on 17 August were higher than those on the other days. In Turpan, the water requirement of an apple tree showed a similar trend with Lintong, and the difference was that the diurnal variation range of water requirement in Turpan was larger than that in Lintong. The reason may be that the diurnal temperature difference and solar radiation changed greatly.

It can be assumed that the plant's water requirement was constant at night since the change in thermal environment was not obvious. Through calculation, the nighttime water requirements of the plant in Lintong were 0.024 L/h, 0.015 L/h, and 0.022 L/h, respectively, and in Turpan they were 0.019 L/h, 0.028 L/h, and 0.032 L/h. Therefore, the monthly plant water requirements of the plant for the two places can be calculated as shown in Figure 15.

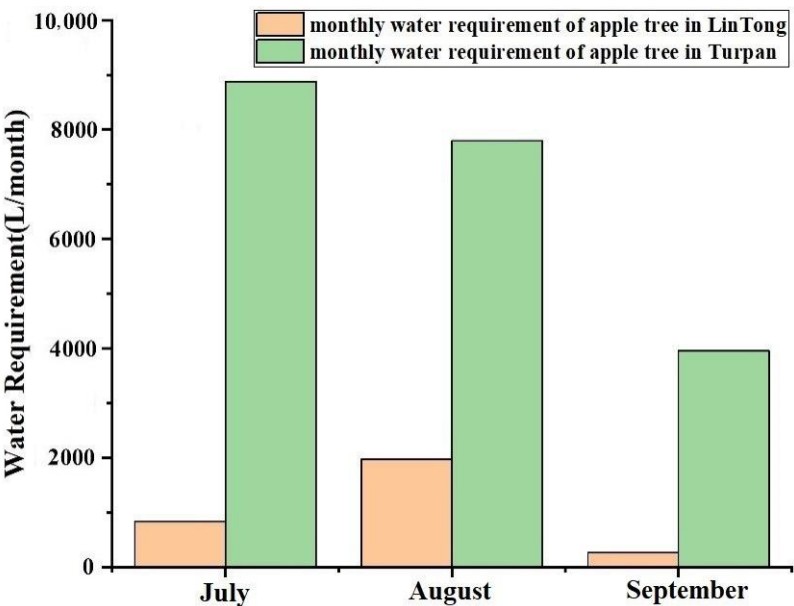

**Figure 15.** Monthly water requirement of the apple tree in Lintong and Turpan.

According to historical data, the monthly rainfall in Lintong was about 20 L in summer, and that in Turpan was about 7 L, which is far from meeting the simulated water requirement of the apple tree. Therefore, the theoretical irrigation amounts required in Lintong were about 820 L, 1960 L, and 250 L per month, and those in Turpan were about 8860 L, 7800 L, and 3960 L per month, respectively.

There have been studies that showed that plants can regulate the surrounding thermal environment because of transpiration. The cooling effect of plants can be calculated by the following equation

$$Q = m_{H_2O} \times l \times 4.18 \tag{25}$$

where $m_{H_2O}$ is the mass of water consumed by plant transpiration, (kg); $l$ is the evaporation heat consumption coefficient of water, and the calculation formula of the evaporation heat consumption coefficient of water is as follows

$$l = 597 - 0.57 \cdot T_{air} \tag{26}$$

The theoretical cooling capacity of plant transpiration on its surrounding environment is

$$\Delta T = Q/\rho_C \tag{27}$$

among them, $\rho_C$ is the volume specific heat capacity of air.

After calculation, the monthly average values of the plant cooling effect in Lintong were 1.75 °C, 2.55 °C, and 1.27 °C, respectively, and the values in Turpan were 9.62 °C, 7.79 °C, and 4.38 °C. It can be seen that plants have a good cooling effect on the environment. Moreover, when a plant is under severe thermal conditions, the cooling effect could be more powerful.

## 5. Conclusions

Formulating a scientific irrigation strategy is an important step to realizing the rational distribution of water resources, especially in the era of an increasing scarcity of water resources. The effect of the thermal environment on plants' water mechanisms and requirements was studied. Results showed that the anatomical features of xylem and the physical properties of sap may affect the plant–water relationship, and plants can lose vitality because of severe thermal environments. Therefore, the stem water delivery capacity and the impact of the thermal environment on plants should be considered when formulating an irrigation strategy. These findings contribute to the formulation of a plant irrigation strategy, especially in hot or arid areas.

**Author Contributions:** Conceptualization, H.L. and H.S.; methodology, H.S.; software, H.L.; validation, J.L.; data curation, H.L. and H.S.; writing—original draft preparation, H.L. and H.S.; writing—review and editing, H.L. and H.S.; supervision, H.S.; project administration, H.S.; funding acquisition, H.S. All authors have read and agreed to the published version of the manuscript.

**Funding:** This work was supported by the National Natural Science Foundation of China (52108099) and research funding of Hunan Provincial Department of Education and Xiangtan University (KZ08051).

**Institutional Review Board Statement:** The study did not require ethical approval.

**Informed Consent Statement:** Informed consent was obtained from all subjects involved in the study.

**Data Availability Statement:** Not applicable.

**Conflicts of Interest:** The authors declare no conflict of interest.

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
