# Peer review of "Analysis of Plant Water Transport Mechanism and Water Requirement for Growth Based on the Effect of Thermal Environment"

_forests, doi:10.3390/f13040583_

Round 1

Reviewer 1 Report

The manuscript has been significantly improved. The authors have added a 'Materials and methods' section, which sufficiently characterises the plant material and explains how to obtain data for subsequent analyses. Sections have been added describing the statistical methods used in the study, including error types for determining deviations of data from the mean value and correlation coefficients. Results of calculations of the parameters mentioned are presented. 
The 'Analysis and discussion' section has also been modified.
The changes applied greatly influenced the level of the manuscript.

Author Response

Dear reviewer:

We very much appreciate your constructive comments and suggestions. We have carefully treated your kind suggestions and revised portions are marked in red in the revised manuscript. We have also checked and corrected the spelling and grammar of the full text carefully. The revised portions are marked in the revised manuscript.

Reviewer 2 Report

I have reviewed this manuscript previously and my main criticisms were:

1-The water use model presented in the paper doesn’t represent any new biological/physical process or have any real novelty that could warrant a publication by itself.

2- The model is overparametrized. The model uses 9 empirical parameters. There are stomatal/plant hydraulic models out there that model plant transpiration responses to climate (including soil moisture responses) with 3 (or less) empirical parameters (e.g., Venturas et al 2018; Tuzet et al 2003). I don’t see any real justification for this excessive number of empirical parameters considering the basic processes being represented in the model. The lack of comparison with other models makes this even more unconvincing.

3- The model isn’t used to produce any novel scientific findings or respond a significant question. The relationship between water use and temperature and humidity seem to just reflect stomatal responses to atmospheric vapor pressure deficit which have been known for decades (Ball et al 1987).

4- The manuscript was poorly organized and poorly written.

The only main criticism the authors have resolved was the writing, which was improved in the revised version even though I still find the manuscript organization very confusing and different from any other plant modelling study I know of. However, unfortunately all the other points remain unresolved. Additionally, I received no direct responses by the authors to any of my comments on the original manuscript or even a track-changed version of the revised manuscript.

References:

Ball, J. T., Woodrow, I. E., & Berry, J. A. 1987. A Model Predicting Stomatal Conductance and its Contribution to the Control of Photosynthesis under Different Environmental Conditions. Progress in Photosynthesis Research, 221–224. doi:10.1007/978-94-017-0519-6_48

Tuzet A, Perrier A, Leuning R. 2003. A coupled model of stomatal conductance, photosynthesis and transpiration. Plant, Cell & Environment 26: 1097–1116.

Venturas MD, Sperry JS, Love DM, Frehner EH, Allred MG, Wang Y, Anderegg WR. 2018. A stomatal control model based on optimization of carbon gain versus hydraulic risk predicts aspen sapling responses to drought. New Phytologist 220:836-50.

Author Response

Dear reviewer:

We very much appreciate your constructive comments and suggestions. We have carefully treated your kind suggestions and revised portions are marked in the revised manuscript. Followings are the reply to each comment.

Point 1: The water use model presented in the paper doesn’t represent any new biological/physical process or have any real novelty that could warrant a publication by itself.

Reply

Thanks for your comment. We have read your comments very carefully again, and we have realized that the application potential of this article was not clearly stated in the submitted manuscript. Based on the effects of thermal environment on plant-water relation, we established the plant water requirement model, optimized the stem water delivery model, and summarized the root water uptake model. Based on the climatic characteristics of hot or arid areas, the limiting effect of stem water delivery capacity on plant was analyzed. Compared with previous studies, the model established in this paper could describe the plant water transport process macroscopically, and may reflect the effect of thermal environment on plant water requirement in some other aspects.

We have further analyzed the limiting effect of stem to illustrate the application potential of the model. The supplemented text has been supplemented in the correspond place of the revised manuscript. (Line number 415-425, with marks version)

In the “Discussion” section, we have supplemented a final paragraph to compare the differences between the previous models and the plant water requirement model to illustrate the model’s application potential. The final paragraph has been supplemented in the corresponding place of the revised manuscript. (Line number 426-444, with marks version)

The supplemented final paragraph is as follows.

“These findings could provide a theoretical basis for the regulation strategy of the agricultural environment. When stomatal regulation cannot regulate the plant-water relation effectively, or stem water delivery capacity is insufficient, the effect of appropriate regulation of the air temperature may be positive. In previous studies, scholars have deeply analyzed the response mechanism of plant to environmental factors (Venturas et al., 2018; Tuzet et al., 2003). Based on these great findings, the plant water requirement model was established in this paper, the stem water delivery model was optimized, and the root water uptake model was summarized based on thermal environment. In order to reflect the relation between thermal environment and plant water requirement, the stomatal model and root water uptake model were established based on thermal environment factors. However, the factors affecting the stomatal movement and root water uptake are very complex, the empirical parameters would be essential to modify these models. Furthermore, it could be found from the discussion made above that severe thermal environment may lead to the loss of plant vitality, and the limiting effect of stem may be an important reason for this phenomenon. Therefore, compared with previous studies, the model established in this paper could describe the plant water transport process macroscopically, and may reflect the effect of thermal environment on plant water requirement in some other aspects.”

Point 2: The model is overparametrized. The model uses 9 empirical parameters. There are stomatal/plant hydraulic models out there that model plant transpiration responses to climate (including soil moisture responses) with 3 (or less) empirical parameters (e.g., Venturas et al 2018; Tuzet et al 2003). I don’t see any real justification for this excessive number of empirical parameters considering the basic processes being represented in the model. The lack of comparison with other models makes this even more unconvincing.

 Reply

Thanks for your comment. We have read the references you mentioned very carefully. These good references have been of a great help to our research. We have supplemented the references (Venturas et al., 2018; Tuzet et al., 2003) in the corresponding place of the revised manuscript. (Line number 431-432, with marks version)

In order to reflect the relation between thermal environment and plant water requirement, we adopted the stomatal model and root water uptake model based on thermal environment factors. However, we found that the factors affecting stomatal resistance and root water uptake rate very complex, therefore we considered to use the empirical parameters to modify these models. We have explained the reasons for using so many empirical parameters in the revised manuscript. The reasons have been supplemented in the corresponding place of the revised manuscript. (Line number 159-163, line number 227-230, with marks version)

We have also calculated the correlation coefficients of these models in the section of “Validation”. The results showed that the results of the model were in good agreement with the experimental results. (Line number 298-303, with marks version)

In the “Discussion” section, we have also supplemented a final paragraph to compare the differences between the previous models and the plant water requirement model to further illustrate the application value of this paper. (Line number 426-444, with marks version)

Point 3: The model isn’t used to produce any novel scientific findings or respond a significant question. The relationship between water use and temperature and humidity seem to just reflect stomatal responses to atmospheric vapor pressure deficit which have been known for decades (Ball et al 1987).

 Reply

Thanks for your comment. We have very carefully read the reference you mentioned and the subsequent relevant references. These good references have been of great help to our research. We have supplemented these references (Ball et al 1987; Sperry J S 2016) and the latest research progress in the corresponding place of the revised manuscript. (Line number 159-163, with marks version)

We adopted the stomatal model based on thermal environment factors to better reflect the effects of thermal environment on plant. We have supplemented some acknowledgement in the corresponding place of the revised manuscript, indicating that not all the calculation methods involved in the stomatal model are entirely new. (Line number 159-163 with marks version)

The supplemented acknowledgement is as follows.

“Research showed (Feddes et al., 2001) that these empirical parameters in the above equations played important roles in modelling the plant water requirement. In addition, it is not an entirely new finding, there were studies (Ball et al., 1987; Sperry JS et al., 2016) provided additional validation that the stomata also do responses to factors other than those in the above equations.”

Point 4: The manuscript was poorly organized and poorly written.

 Reply

Thanks for your comment. We have reorganized the structure of the paper. The spelling and grammar of the full text have been checked and corrected carefully.

Reviewer 3 Report

Dear Author

I have read the revised manuscript (Forest -1656217). Entitle: Analysis of Plant Water Transport Mechanism and Water Requirement for Growth Based on the Effect of Thermal Environment for publication of Forest MDPI. This is the second submission made by the author. I review the original manuscript already. The author addressed all the questions and suggestions that I raised the issue in the review of the original manuscript. I satisfy the author’s revisions throughout the paper. Especially author improved the introduction and discussion section very well inflow. The abstract issue is also solved by the author. Now, this manuscript improved the flow of writing, which was comparatively shallow in the original version but in this revised copy author addressed all the quarries and suggestions very well. Before accepting this manuscript if there is anything needed to be revised by the author, especially English grammar, or spell check, I request this manuscript is currently in “Minor Revision” and any grammatical error author may improve in this stage. Thank you.

Author Response

Dear reviewer:

We very much appreciate your constructive comments and suggestions. We have carefully treated your kind suggestions and revised portions are marked in the revised manuscript. Followings are the reply to your comment.

Point 1: Before accepting this manuscript if there is anything needed to be revised by the author, especially English grammar, or spell check, I request this manuscript is currently in “Minor Revision” and any grammatical error author may improve in this stage.

Reply

Thanks for your comment. The spelling and grammar of the full text have been checked and corrected carefully.

This manuscript is a resubmission of an earlier submission. The following is a list of the peer review reports and author responses from that submission.

Round 1

Reviewer 1 Report

Review of the manuscript entitled "Analysis of water transport mechanism and water requirement of plants for growth based on the effect of thermal environment" by Haolin Lu, Hongfa Sun, and Jibo Long.

The manuscript by Hongfa Sun (corresponding author) and others aims to present a model to evaluate the minimum water requirements of plants under normal growth conditions. The authors presented several initial models based on which the model evaluated was developed.

In the "Introduction" section of the manuscript, the Authors properly present the problem and importance of water balance, the factors affecting the proper water balance of plants, and the assumptions of the component models.

The biggest doubt is the lack of a "Materials and methods" section of the manuscript describing the methods used to obtain data to create the model. Although the authors state that the analyses are based on measurements made by Gong in 2005, they do not describe the methodology of making these measurements (lines 265-269). The reference list shows that the results of this research are not published in a scientific article. They state that transpiration coefficients were calculated based on these data and the relevant equations. Therefore, nothing is known about the plants and their growth conditions at the time of the measurements made in 2005. At the same time, the Authors state that a comparison between the simulation values and actual values is presented in Figure 6. How the actual values were obtained (lines 299-302). The methodology of the measurements is not presented. The statistical methods used to compare these data series also are not presented.

In the Results and Discussion section of the manuscript, the authors again refer to previously obtained data. Which results do the Authors have in mind - is it the data obtained by Gong (2005) or some other data. How they were obtained.

The "Application and Analysis" section of the manuscript presents the results of measurements at two locations. Again, the question arises: how were these data obtained, what are the characteristics of the study plots, what statistical methods were used.

In all figures, only averages are presented. There is no data to determine the deviation of the obtained data from the mean, for example SD or SE.

The work is relevant to the physiology of plant water balance. However, it contains serious deficiencies that require major corrections and additions.

Reviewer 2 Report

REVIEW

ANALYSIS OF PLANT WATER TRANSPORT MECHANISM AND WATER REQUIREMENT FOR GROWTH BASED ON THE EFFECT OF THERMAL ENVIRONMENT

Overall view

The authors use known equations to build a mechanistic model of water absorption and transport according to the water requirements of plants. They use an apple tree as the target plant and analyze the limiting effects of stem water delivery capacity and root water uptake capacity on canopy transpiration under hot season climate conditions in rainy and arid areas. Using published data, the authors conclude that the theoretical results have good accuracy in calculating the water requirement of the tree. Among the environmental parameters affecting transpiration rate, the air temperature was found to be the most important factor, and the wind velocity had little effect on the water requirement. The authors appear to confound transpiration with evapotranspiration in their literature review.

For the calculation of transpiration (water requirement) the authors use the leaf area of the tree and transpiration as the driving force of water uptake and transport. They use Rijtema´s model of stomata functioning but they do not use a stomata response to plant internal physiological variables and soil dryness.  The results agree with what is known about plant water responses to environmental variables

Specific comments

Lines 33 34. By definition wilting is caused by lack of water either directly or indirectly.

Lines 40 41 English

Line 56 “transpiration rate” Do you mean evapotranspiration?

Lines 58 61 , “but the coordination between stem water carrying capacity and root water  uptake capacity and transpiration rate is often ignored, that is, when the plant transpiration rate is large, the stem water delivery rate and root water uptake rate may not meet  the water requirement of transpiration.”    Using P-M equation?

Lines 61 62 English

Line 63 “catheter” Do you mean xylem conducting units?

Line 103 English

Lines 105 106 “to accurately measure the canopy resistance, which is theoretically affected by the stomatal resistance of each leaf” You are confounding stomata resistance with canopy resistance. Please take a look to the derivation of P-M equation.

Line 122 change steam by vapor

Lines 126-163, equations not checked by me.

Line 172 Equation 14 should overestimate tree transpiration. How do you correct LAI?

Lines 179 181 “In this paper, assuming that the diameter of the stem catheter is equal to the stem diameter of the plant, ignoring the thickness of the stem sapwood, the water transfer rate of the plant stem” There is something wrong here or poor English. Please check.

Line 183. This is Poiseuille´s law. How do you integrate among stem vessels? Or you use another indicator?

Lines 202-203 “However, when the water delivery performance of the stem meets the water requirement of the plant and the plant grows normally” What do you mean by “normally”, and water requirement of the plant? Are you implying that transpiration is determined by leaf transpiration only, as indicated in any plant-water relations textbook?

Line 212 Please indicate how do you obtain all those coefficients: “Where a1, b1, b2, b3, c1, c2 are empirical coefficient”

Line 216 Q3= K2 qAnother fudge factor

Line 223 What is the k1 factor appearing in eq 19?

Line 227 What do you mean by water requirement of the plants?

Line 229  I do not understand the origin of eq 20 : Q= K2 qr

Lines 231 233. I do not understand here; I would assume that at this point the stomata are closed and the “plant water requirements” are equal to the cuticular transpiration (not even mentioned by the authors)

Line 271 Full text required

Lines 278 297 You do not need to put all these numbers in the text as we can read the graphs. If you wish arrange a table

Line 304  Graph legend should be complete

Lines 305 326 You can arrange a table with these results

Line 309  Actual or maximum?

Line 312 What do you really mean by actual?

Line 305 326 Could arrange in a table

Line 329 330 ; solar radiation, air temperature, air humidity and wind velocity. This is obviously trivial

Line 343 Legends in full are required

Line 356 The major physiological characteristics are located at the stomata level. Not addressed by the authors

Line 380 Full legend required

Lines 383 386. I believe your explanation is wrong: “The reason may be the influence of stomatal conductance. When the air temperature is low, the stomatal conductance is smaller, and the air temperature change has little influence on the change of plant water requirement” Please take a look to what happens with DPV and temperature for a fixed stomatal opening.

Lines 390 391 English. I do not understand

Line 403 “with the increase of relative humidity, the stomatal conductance decreased” As I understand what happens is the opposite

Line 450 English. What do you mean by water requirement variation law?

Line 483. Full legends required for each table/figure

Lines 492 493  English

Line 526  Full legends require

Conclusion

This manuscript needs to be reviewed by a native English speaker. The manuscript as it stands appears to be solely a computing exercise. The authors should underscore what is really new in their manuscript. I strongly advise the authors to include a plant physiologist in their research team. The authors need to fully understand the role of stomata in plant transpiration and the internal, physiological factors affecting their response.

Reviewer 3 Report

General comments

I have read the manuscript (Forest -1609683). Entitle Analysis of Plant Water Transport Mechanism and Water Re- 2 requirement for Growth Based on the Effect of Thermal Environment Haolin Lu et. al., for publication of forest MDPI. In this study, the author investigates the plant water requirement and thermal environment, a calculation model was proposed about water consumption and water transport in plants. The results show good accuracy in calculating the water requirement for normal plants growth. Among the environmental parameters affecting transpiration rate, the air temperature is the most important factor affecting the minimum water requirement of plants growth, and the wind velocity has little effect on the water requirement for plant growth.

The overall research is well conducted, and research is obvious application potential because to determines the water consumption rate and its requirement according to the weather conditions and the main factors that determine the plant water relation. In this sense, the manuscript is much valuable. However, I found some points, especially the flow of the text is not smooth and sometimes I found the shallow writing and lack of potential references, and lack of connection of story in different paragraphs, especially in the introduction and discussion sections. In discussion, the author should be deal with the physiological traits and plant water relation in-depth and those should relate to the biological perspectives. I found the lack of potential and appropriate references to support the findings. The author should provide examples and enough references to support their findings. Some of the references I mention below that help to improve this manuscript quality better than before. Overall after I evaluate this manuscript, I request the author for the “MAJOR REVISION” and, I request to authors for revision according to the rules of the journal and correct the bibliography.

 Major suggestions

1) Abstract Issue: The author wrote the important finding in their abstract, but the text seems confusing mainly due to less clarity of the main results and their theme in the abstract section. Author should clarify Line no. 17-21 more clarity, it is not enough that the “environmental factors affecting the transpiration and air temperature are important factors” it is already well known but author should present their result in more robust form, not the just statement. I think the abstract should be revised significantly by the author. Please remember that the abstract should more logical, short, concise, and informative. Your abstract should reflect your study and major findings while shortly observed by readers. Please make the necessary corrections and cross-check the word limit of your abstract.

2) Introduction: The author will describe the starting of the introduction with good background and well include the subject matter of the global warming issue and its negative effect in the near future prediction which is appreciated. However, the author should be improved the introduction section further. Especially author should deal the plant water relation as well as the author should to deal with the dealing the stomatal conductance (gs). Please see Line no. 61-63 where the author tried to deal with stoma closing and opening and its effect on the plant water relations and water use efficiency.  The author should be furthermore clarified and mention more text that the stomatal conductance (gs) opening and closing and related activities to control the water movement. I request to author to connect this part by dealing the stomatal controlling gs traits and carbon isotope by the referencing of these two articles which are good and latest references for the discussion section. https://doi.org/10.1016/j.scitotenv.2021.146466 Title: Evaluation of morphological, physiological, and biochemical… and (2) doi: 10.3390/plants8070232. “WUE is specified species and its controlled by the stomata opening and closing because stromal full opening and partial opening of stomata partial closing cause the turgid the plant part”. Moreover, it may be varying the transpiration rate from the leaf canopy level, this story author should be mentioned by citing the above two kinds of literature to improve the text more clarity.

3) Ultimate goal and research hypothesis: The author should further improve the research hypothesis and objective in the last paragraph in the introduction (Line from 75-83). Furthermore, the author should present the add the goal of the study alone with the objectives. Accordingly, the author presents the hypothesis based on the main practical application of the study about and why It is an important prerequisite for the development of plant carbon storage technology to develop appropriate calculation models of plant growth water consumption and water transport. The author’s hypothesis should be clear according to the main approach of the research. The hypothesis should be very clear in the introduction sections because, without appropriate literature, questions, or hypotheses in the introduction section the entire objective of the manuscript is unclear, please make the necessary correction in this section.

4) Discussion sections: Improve your discussion more logically, author should be properly covering the interpretation of the all parts in the discussion sections. I saw the author include a lot of good parameters and literature but the “plant water-relation, gs role and to conserve water and its related story is comparatively weaker. Heat and drought showed similar symptoms for the plant both causing loss the vitality of the plant and its hydraulic conductivity by lower the leaf water status and whole plant-water relation. In the discussion, author should present this issue clearly by adding literatures that directly match with the plant water relations and its affecting factor such as stem vessel structures. This literature Entitle: Impact of drought stress on photosynthesis responses, leaf water….. “DOI:10.1016/j.scienta.2018.11.021 is good reference to interpretation to the plant water relation for the discussion section. The point “Plant water relation and leaf water status, or sap movement of the plant depend on the xylem anatomical features such as vessel diameter, vessel number, vessel size” this part should include in discussion section.

Others Minor suggestions

  1) Improving the writing in the Line no 40, 69, and 70 and mentioning the space while making the bracket.  Similarly, please be careful of this error throughout the manuscript.  

2) Line no. 271: Please mention the detailed description related to the figure element in figure 2, Please see this error in others figures too because the legend should be full of information.

3) Line no. 282-297 Please make this section more concise and only mention the very important numerical data or some case author may keep in the range.

4) Line no. 304: Please improve the legend of figures 6 and 9, 10, 11, and others remain, please remember that the legend should include full information.

5) Conclusion section (Line no. 582): Please improve the conclusion section, which should be in good glow with include all necessary components of the study and does not repeat the result section. Conclusions should be present the future insight of the research based on your current finding and the strength of your results for the future research guideline. The conclusion for me comes off as repetitive of the abstract or a summary of the results section. I would love to read striking points and take-home messages that will linger in the readers’ minds. What is the novelty, how does the study elucidate some questions along this field, and the contributions the paper may offer to the scientific community?

6) Reference: (Line no. 610): lease double-checks the citations, their style, spell check, and other grammatical errors. moreover, I request to authors for revision throughout the manuscript according to the journal rules.

Good Luck !

Reviewer 4 Report

This manuscript describes a model to simulate plant water use, evaluates the model against sap flux data from apple tree and later uses some type of sensitivity analysis to investigate which environmental factors control the modelled transpiration. The model has a good fit to the sap flow observations and the authors conclude the transpiration rate is mostly controlled by temperature.

This manuscript does not have any novel or relevant finding in my opinion. The model itself does not contain any significant scientific advancement or process that haven’t been represented before in other models. In fact, the model omits much of the recent advances in plant hydraulic modelling on its representation of plant water transport and stomatal regulation (Sperry & Love, 2015; Sperry et al 2017). Additionally, the model requires several empirical parameters of difficult physical or biological interpretation which make it difficult to be used in other studies and applications, such as ecosystem/climate models. The model fits the observations well, but that does not say much about its usefulness considering the number of empirical parameters the model has that can be calibrated/fitted to the data.  A comparison with other existing water transport models and/or alternative model formulations would be essential to really show the value of the proposed model.

The model predicts that plants use more water during hot climates which is reasonable but not particularly new information about plant functioning. Even this effect is not very well explained in the paper, as this higher water use could be caused by the temperature effect on VPD (Eq 10), its effect on stomatal closure (Eq 11), leaf temperature (Eq 4) or something else.

Additionally, the paper is difficult to follow mainly due to a very unorthodox organization (results are spread through several different sections) and poor English. I recommend the authors to thoroughly revise the English and restructure their paper following the standard Introduction, Methods (which might include model description), Results and Discussion (which should include the model validation results) organization. The introduction (and other sections of the paper) is also unreasonably focused on growth and carbon storage, even though the model does not really simulate any of these processes.

Specific comments:

L141: The stomata also responds to soil moisture changes. Why isn’t this effect included in the model? If the model was designed to be run in arid areas as you state in several parts of the text, it should include this effect.

L183: Xylem capacity to transport water declines as transpiration (and consequently the water tension gradient) increases, see Sperry & Love (2015) for a recent model of this process.  Why was this basic process ignored in the model?

L211: Please explain the rationale of this empirical equation, how root water uptake depends on wind velocity? Why root water uptake depends on air humidity instead of soil humity?

L237: How deltaP and deltaPmax are calculated?

L268: Specify which method was used to measure sap flux, there are several thermometric methods to measure sap flux (heat ratio method, thermal dissipation, heat field deformation, etc.)

L278: There is no need to repeat the values that are shown in the plot.

L299: How model parameters were obtained? Your model depends on several empirical coefficients, was the model fitted to the data? Please describe how the model fitting (or calibration) was performed and provide a table with the values used for every model parameter.

L322: Besides the error provide some correlation statistics, like the r or the r2 between predictions and observations.

References:

Sperry JS, Love DM. 2015. What plant hydraulics can tell us about responses to climate‐change droughts. New Phytologist. 207(1):14-27.

Sperry JS, Venturas MD, Anderegg WR, Mencuccini M, Mackay DS, Wang Y, Love DM. 2017. Predicting stomatal responses to the environment from the optimization of photosynthetic gain and hydraulic cost. Plant, cell & environment. 40(6):816-30.